# Insight into Tetramolecular DNA G-Quadruplexes Associated with ALS and FTLD: Cation Interactions and Formation of Higher-Ordered Structure

**DOI:** 10.3390/ijms241713437

**Published:** 2023-08-30

**Authors:** Matja Zalar, Baifan Wang, Janez Plavec, Primož Šket

**Affiliations:** 1Slovenian NMR Center, National Institute of Chemistry, Hajdrihova 19, SI-1000 Ljubljana, Slovenia; matja.zalar1@um.si (M.Z.); baifan_wang@outlook.com (B.W.); janez.plavec@ki.si (J.P.); 2EN-FIST Center of Excellence, Trg OF 13, SI-1000 Ljubljana, Slovenia; 3Faculty of Chemistry and Chemical Technology, University of Ljubljana, Večna pot 113, SI-1000 Ljubljana, Slovenia

**Keywords:** ALS, FTLD, c9orf7, DNA, quadruplex, NMR, cations

## Abstract

The G_4_C_2_ hexanucleotide repeat expansion in the *c9orf72* gene is a major genetic cause of familial amyotrophic lateral sclerosis (ALS) and frontotemporal lobar degeneration (FTLD), with the formation of G-quadruplexes directly linked to the development of these diseases. Cations play a crucial role in the formation and structure of G-quadruplexes. In this study, we investigated the impact of biologically relevant potassium ions on G-quadruplex structures and utilized ^15^N-labeled ammonium cations as a substitute for K^+^ ions to gain further insights into cation binding and exchange dynamics. Through nuclear magnetic resonance spectroscopy and molecular dynamics simulations, we demonstrate that the single d(G_4_C_2_) repeat, in the presence of ^15^NH_4_^+^ ions, adopts a tetramolecular G-quadruplex with an all-*syn* quartet at the 5′-end. The movement of ^15^NH_4_^+^ ions through the central channel of the G-quadruplex, as well as to the bulk solution, is governed by the vacant cation binding site, in addition to the all-*syn* quartet at the 5′-end. Furthermore, the addition of K^+^ ions to G-quadruplexes folded in the presence of ^15^NH_4_^+^ ions induces stacking of G-quadruplexes via their 5′-end G-quartets, leading to the formation of stable higher-ordered species.

## 1. Introduction

Guanine-rich nucleic acid segments have the ability to form non-canonical, four-stranded G-quadruplex structures, which consist of at least two G-quartets. These structures are planar arrays of four guanines held together by eight Hoogsteen hydrogen bonds [1,2,3]. In addition to G-quadruplexes, G-rich sequences can also adopt other tetrahelical arrangements [4,5].

Within the human genome, G-rich sequences are predominantly found in telomeres [6] and promoter regions of genes [7]. This suggests that G-quadruplex formation may play a role in gene regulation and expression [8,9,10,11]. The expansion of the G_4_C_2_ hexanucleotide repeat in the *c9orf72* gene has been identified as the primary genetic cause of familial amyotrophic lateral sclerosis (ALS) and frontotemporal lobar degeneration (FTLD) [12]. Although the exact disease mechanism is still under investigation, three hypotheses have been proposed: RNA toxicity, accumulation of dipeptide repeat proteins, and *c9orf72* haploinsufficiency. Despite their differences, all these mechanisms are related to the formation of stable structural features by the expanded G_4_C_2_ repeats [13].

Previous studies have reported that both RNA [14,15,16] and DNA [17,18,19] can form various types of G-quadruplexes depending on the number of G_4_C_2_ hexanucleotide repeats.

Recent research has also demonstrated a correlation between G-quadruplex formation and the development of ALS and FTLD in mice [20,21]. It is also known that *C9orf72* hexanucleotide repeat expansion interacts with several protein partners, such as DAXX, which drives reorganization of genomic structures and causes epigenetic changes in patient cells [22]. Furthermore, it was shown that RNA G_4_C_2_ repeats interact with the splicing factor hnRNP H, which leads to elevated insoluble aggregates in patient brains [23], while binding of DEAH-box helicase 36 resolves RNA G_4_C_2_ G-quadruplexes and as such is essential in regulating the repeat-associated non-AUG translation [24,25]. Furthermore, C-rich antisense DNA d(G_2_C_4_) and RNA (G_2_C_4_) repeats can form stable non-canonical structures, such as i-motifs and protonated hairpins [16,26,27], which adds another layer of complexity to the association between *c9orf72* and ALS/FTLD. Additionally, ions that are present in the cells at different concentrations may also influence the formation of DNA/RNA structures.

G-quadruplexes can differ in the number of G-quartets, strand stoichiometry and their progression, types of loops, and loop orientation [28]. Furthermore, a recent study showed that even a single mutation in the G-tract prevents formation of G-quadruplexes [29]. Unimolecular G-quadruplexes formed by d(G_2_C_4_)_n_ repeats exhibit a high degree of structural polymorphism in solution, and sequence modifications are typically needed to reduce the number of structures formed in solution [19]. The topology and thermal stability of G-quadruplexes are known to be significantly influenced by cations [30,31,32,33,34,35,36]. Cations play a crucial role in G-quadruplex formation by reducing repulsion between guanine carbonyl oxygen atoms and promoting the stacking of G-quartets [37]. Among the various cations that promote G-quadruplex formation, K^+^ ions are the most effective inducers, followed by NH_4_^+^ and Na^+^ ions [38,39]. Cations are typically located between two G-quartets, within the plane of a G-quartet, or between the outer G-quartet and loop residues [40]. The specific cation binding site in G-quadruplexes is influenced by factors such as the *syn-* and *anti*-orientation of guanine residues in a G-quartet. Additionally, cations are not static and can move between binding sites and exchange with cations in the bulk solution [41,42,43,44,45,46]. ^15^NH_4_^+^ ions have been utilized in nuclear magnetic resonance spectroscopy (NMR) studies as a non-metallic substitute for K^+^ ions, enabling the identification of cation binding sites and the study of their exchange kinetics [41,42,43,44,45,46,47]. Previous research has demonstrated that the strand stoichiometry and loop topology influence the kinetics of cation movement [44,45,46,48]. Furthermore, the presence of *syn*-quartets significantly decreases the rate of cation exchange [45,49].

In this study, our primary interest was to investigate the interaction between different cations and the G-quadruplex structure formed by the short hexanucleotide repeat d(G_4_C_2_), where a single repeat was chosen to eliminate the structural polymorphism that prevents the analysis of cation binding sites within G-quadruplexes formed by longer (G_4_C_2_)_n_ repeats. We expected that the mentioned oligonucleotide, in the presence of ^15^NH_4_^+^ ions, would form a tetramolecular G-quadruplex, which could have a parallel or antiparallel orientation of strands. In the former case, G-quartets would be formed, and possibly C-quartets, while in the case of antiparallel strands, we would expect the presence of GC-quartets or GC base pairs in addition to G-quartets. The localization of cations would depend on the resulting 3D structure, and it may not be restricted to residing solely between the G-quartets. We also expected certain binding sites to exhibit a preference for cation binding, leading to their higher occupancy. Additionally, we anticipated that the kinetics of cation exchange between individual binding sites within the structure and the cations from the bulk solution would be strongly influenced by the structural elements of the formed structure. This could manifest in preferential movement of cations in specific directions. Moreover, we expected that the addition of biologically relevant K^+^ ions would cause the replacement of ^15^NH_4_^+^ ions within the G-quadruplex with K^+^ ions, potentially resulting in changes in the folding topology and, possibly, formation of higher-order structures. In the case of higher-order structure formation, the question arose as to how many ^15^NH_4_^+^ ions need to be replaced by K^+^ ions to achieve such formation.

## 2. Results

### 2.1. d(G_4_C_2_) Forms Tetramolecular G-Quadruplex in the Presence of ^15^NH_4_^+^ Ions

Initially, the folding of d(G_4_C_2_) was assessed using ^1^H NMR spectra. At low cation concentrations, a single set of signals corresponding to aromatic and sugar protons of guanine and cytosine residues was observed, along with a broad signal at approximately δ 10.9 ppm. This signal indicated the formation of GG N1 carbonyl symmetric base pairs in the pre-folded species, while no signals indicating Watson–Crick base pairs were detected (Figure 1A and Appendix A).

Upon addition of 100 mM of ^15^NH_4_Cl, the formation of G-quadruplexes was initiated, and a fully folded state was achieved after three weeks (Figure 1B, Appendix A). Four distinct signals of imino protons from guanine residues indicated that all guanines in d(G_4_C_2_) were involved in Hoogsteen-type hydrogen bonds. No signals suggesting the formation of hemi-protonated C^+^-C base pairs were observed, even at pH 4.5 and 5 °C (Appendix A). The NMR data obtained from various 2D NMR experiments (Appendix A) consistently supported the formation of a parallel tetramolecular G-quadruplex with a C_4_ symmetry axis and four G-quartets (Figure 2A), with G1 having a *syn*- and G2–G4 an *anti*-conformation of their glycosidic torsion angle (Appendix A). The size and strand orientation of the G-quadruplex structure were supported by the translation diffusion coefficient (D_t_) of 1.41 ± 0.05 × 10^−6^ cm^2^ s^−1^ and the CD spectrum (Appendix A). In the presence of 100 mM ^15^NH_4_Cl, [d(G_4_C_2_)]_4_ also exhibits a remarkable thermodynamic stability, with unfolding occurring only at 83 °C (Appendix A).

### 2.2. All syn G-Quartet Directs Cation Movement between [d(G_4_C_2_)]_4_ G-Quadruplex and Bulk Solution

The 2D ^1^H-^15^N HSQC NMR spectrum provided evidence of three distinct binding sites for ^15^NH_4_^+^ ions within the [d(G_4_C_2_)]_4_ G-quadruplex. These binding sites were found to be located between pairs of adjacent G-quartets, as confirmed by the analysis of 2D NOESY and ROESY spectra (Figure 2 and Appendix A). Binding site O_5_ (δ_H_ 7.27 ppm) corresponds to the binding site between G1 and G2 G-quartets, binding site I (δ_H_ 6.98 ppm) is located between G2 and G3 G-quartets, while binding site O_3_ (δ 7.02 ppm) is located between G3 and G4 G-quartets.The volume integrals ratio of the autocorrelation cross-peaks O_5_, I, and O_3_ was 2:1:1, indicating unequal occupancy of the binding sites (Figure 2B).

A series of 2D ^1^H-^15^N NzExHSQC spectra with mixing times ranging from 13 ms to 3 s was acquired to evaluate the kinetics of ^15^NH_4_^+^ ion movement between binding sites within the G-quadruplex and with ions in the bulk solution. Movements of ^15^NH_4_^+^ ions between binding sites along the central core of the [d(G_4_C_2_)]_4_ G-quadruplex were observed as the exchange cross-peaks O_5_I, IO_5_, IO_3_, and O_3_I, while ion movement from binding site O_3_ into the bulk solution was characterized by the exchange cross-peak O_3_B (Figure 2C). No resolved cross-peak corresponding to ^15^NH_4_^+^ ion movement from the bulk solution to binding site O_3_ was observed, possibly due to overlap of a weak cross-peak with autocorrelation cross-peak B. No ion movement between binding site O_5_, characterized by an all-*syn* G-quartet, and the bulk solution was detected. However, the 2D ^1^H-^15^N HzExHSQC experiment showed that protons are able to exchange between ^15^NH_4_^+^ ions at binding site O_5_ and those in the bulk solution (Figure 2C), but the quantitative analysis was not possible due to the low peak intensity.

The apparent rate constants of cation exchange were obtained by fitting the integrals of autocorrelation and exchange cross-peaks (Appendix A, Appendix A) and were in the order of 2 s^−1^ or slower. The fastest exchange was observed for movement of ions from O_3_ to I, and from O_3_ to the bulk solution. The observed differences in the apparent rate constants imply the existence of temporarily vacant cation binding sites within [d(G_4_C_2_)]_4_, which is in agreement with the different binding site occupancies observed in the 2D ^1^H-^15^N HSQC spectrum.

### 2.3. C5 Residues Form a Well-Defined C-Quartet within [d(G_4_C_2_)]_4_ G-Quadruplex

A set of G-quadruplex structures formed by the d(G_4_C_2_) sequence in the presence of ^15^NH_4_^+^ ions was calculated using NMR restraints (Appendix A). Ten final structures were selected based on their lowest energy and the smallest number of violations of experimentally derived structural restraints. The resulting structure that best fit all the experimental restraints was a tetramolecular, parallel G-quadruplex with four G-quartets (ID 8C7B, Figure 3A). In Figure 3A, the 5′-end G-quartet is formed by G1 guanine residues that adopt a *syn*-orientation around the glycosidic bond, while all the other G-quartets consist of guanine residues in the *anti-*conformation. The hydrogen bonds within the all-*syn* and all-*anti* G-quartets have anticlockwise and clockwise orientations, respectively. The well-defined core of the G-quadruplex has a pairwise heavy-atom RMSD of 0.80 Å. The 5-member rings of G1 and G2 residues overlap, which agrees with the *syn–anti* sequential step. Stacking interactions between the G2–G3 and G3–G4 residues occur through the partial overlap of both the five- and six-member rings of guanine residues, as expected for the *anti–anti* sequential step (Figure 3A). All four grooves have a medium width, as all the guanine residues within each individual G-quartet exhibit the same *anti-* or *syn*-orientation around the glycosidic bonds. The C5 residues form a well-defined C-quartet, stabilized by hydrogen bonds in the amino-O2 geometry, that overlaps with the 6-member ring of G4 residues (Figure 3A) and the presence of the cations. The well-defined C-quartet is reflected in a slight increase in RMSD to 0.97 Å compared to the RMSD obtained for the G-quartet core. The C6 residues do not exhibit any stacking interactions.

### 2.4. Vacant Cation Binding Sites Impede Cation Movement within [d(G_4_C_2_)]_4_ G-Quadruplex

Since NMR experiments showed that binding sites I and O_3_ had lower occupancy by ^15^NH_4_^+^ ions compared to O_5_, MD simulations were conducted to gain further insights into the localization of ammonium ions. Initially, MD simulations were performed with three NH_4_^+^ ions present at binding sites O_5_, I, and O_3_ within the [d(G_4_C_2_)]_4_ G-quadruplex, with a total cation concentration of 115 mM (Appendix A). After 7 ns of simulation, the NH_4_^+^ ions at binding site O_3_ moved to the cavity between the G-quartets defined by G4 and C5 residues and remained there for the rest of the 200 ns MD simulation (Appendix A). Despite the absence of cations at binding site O_3_, the structure of the [d(G_4_C_2_)]_4_ G-quadruplex remained stable during the MD simulation (Appendix A). This binding site between G4 and C5 residues was designated as O_GC_. Another ^15^NH_4_^+^ ion was trapped between C5 and C6 residues, but due to the flexibility of C6 residues, this cation binding site was not well defined (Appendix A). MD simulations where NH_4_^+^ ions were absent at the beginning of the simulations from binding site O_5_ showed that the NH_4_^+^ ion at binding site I immediately moved to binding site O_5_ (Appendix A) and remained there for the rest of the simulation. In contrast, MD simulations with NH_4_^+^ ions absent at binding sites I or O_3_ demonstrated that, in both cases, the G-quartet core of the [d(G_4_C_2_)]_4_ G-quadruplex remained stable, and cation movement was not observed (Appendix A). The unexpectedly slow cation movements observed for the tetramolecular G-quadruplex, as noted from NMR data, are, therefore, a result of vacant cation binding sites I or O_3_, in addition to the all-*syn* G-quartet at the 5′-end.

The *syn–anti* and *anti–anti* steps within the [d(G_4_C_2_)]_4_ G-quadruplex lead to distinct stacking modes between the corresponding G-quartets. The binding of cations between two G-quartets is directly influenced by the negative charges of the O6 atoms within the G-quartet. Therefore, the alignment of the negative charge centers of the G-quartet core could affect the binding affinity for cations. The geometric parameter d represents the distance between the center of the first G-quartet, e.g., composed of G3 residues, and the projection of the center of the adjacent G-quartet, e.g., composed of G4 residues, onto the plane defined by the first G-quartet (Figure 4). Different values of d influence the octahedral coordination geometries of cations and, consequently, their binding affinity. The values of d between G-quartets consisting of G1 and G2, G2 and G3, and G3 and G4 residues in [d(G_4_C_2_)]_4_ in the presence of ^15^NH_4_^+^ ions are 0.4, 0.5, and 0.6 Å, respectively. Since the most populated binding site, O_5_, is characterized by the smallest d, perfect alignment of the negative charge centers would result in the strongest binding affinity for cations.

### 2.5. Free-Energy Barriers of ^15^NH_4_^+^ Ion Movement Confirm Asymmetry between Binding Sites

NMR experiments revealed that the exchange of ^15^NH_4_^+^ ions from the central channel of the [d(G_4_C_2_)]_4_ G-quadruplex with the bulk solution occurs through the 3′-end. To investigate the free-energy profiles of NH_4_^+^ ion movements, adaptive steered molecular dynamics [50,51] were employed. The potential of mean force (PMF) profile depicted in Figure 5 illustrates that the free-energy barriers for NH_4_^+^ ions to cross the outer G-quartets, consisting of G4 and G1 residues, to the bulk solution were 10.3 and 20.9 kcal/mol, respectively. The significantly higher energy barrier for the G-quartet composed of G1 residues aligns with the NMR observation that ^15^NH_4_^+^ ions do not move between the O_5_ binding site and the bulk solution. The energy barriers for NH_4_^+^ ion movements through G-quartets composed of G2 and G3 residues were approximately 12.2 and 8.7 kcal/mol, respectively.

### 2.6. [d(G_4_C_2_)]_4_ G-Quadruplex Dimerizes into Higher-Order Structures in the Presence of K^+^ Ions

Upon gradual titration of KCl into a solution of [d(G_4_C_2_)]_4_ G-quadruplex, folded in the presence of ^15^NH_4_Cl alone, the imino region of ^1^H NMR spectra exhibited additional signals, indicating the presence of various G-quadruplexes in the solution (Figure 6). At 80 mM of KCl, a single G-quadruplex species was observed again, albeit different from the initial one. NMR and CD data revealed the formation of a parallel tetramolecular G-quadruplex with a C_4_ symmetry axis, with the G1 residue in the *syn*-conformation (Appendix A). Furthermore, the D_t_ value gradually decreased upon the addition of KCl, reaching 1.19 ± 0.05 × 10^−6^ cm^2^ s^−1^ at 80 mM of KCl, indicating the dimerization of two tetramolecular G-quadruplexes, hereafter referred to as the [d(G_4_C_2_)]_4_ G-quadruplex dimer. In a proton–deuterium exchange experiment, the imino protons of G1 and G2 exchanged at a slower rate compared to G3 and G4, suggesting that dimerization occurs through the 5′-end G-quartets (Appendix A). The [d(G_4_C_2_)]_4_ G-quadruplex dimer exhibited significantly higher thermal stability compared to the one in the presence of ^15^NH_4_Cl alone, with no unfolding observed, even at 95 °C (Appendix A).

^1^H-^15^N HSQC experiments conducted at each addition of KCl provided insight into the exchange of ^15^NH_4_^+^ with K^+^ ions at individual binding sites within the G-quadruplex (Figure 6). Combining the measured D_t_ values with the HSQC data revealed that dimerization occurs before ^15^NH_4_^+^ ions fully exchange with K^+^ ions (Figure 7). This further indicated the presence of an additional K^+^ ion at the interface of two tetramolecular G-quadruplex structures (Figure 7). Within a G-quadruplex unit, K^+^ ions initially replaced ^15^NH_4_^+^ ions at the O_3_ binding site. Simultaneously, additional cross-peaks O_5_’ and I’ with ^1^H chemical shifts up-field of O_5_ and I appeared (Figure 6). These cross-peaks indicate the presence of ^15^NH_4_^+^ ions within G-quadruplex species where the O_3_ binding site is occupied by a K^+^ ion (Figure 7). Approximately 70% of ^15^NH_4_^+^ ions at the O_3_ binding site were replaced upon the addition of 1.0 mM of KCl. At 2.0 mM of KCl, around 90% of the O_3_ binding sites and 70% of the I binding sites were occupied by K^+^ ions. The existence of G-quadruplexes with K^+^ ions at the O_3_ and I binding sites and ^15^NH_4_^+^ ions at the O_5_ binding site was evident from the presence of an additional cross-peak in the 2D ^1^H-^15^N HSQC spectrum (O_5_”). At 5.0 mM of KCl, approximately 50% of the G-quadruplexes had all three binding sites occupied by K^+^ ions, while in other structures, ^15^NH_4_^+^ ions still resided at the O_5_ binding site or both the O_5_ and I binding sites. In the presence of 20 mM of KCl, approximately 80% of the G-quadruplexes had all three binding sites occupied by K^+^ ions. At 80 mM of KCl, almost complete replacement of ^15^NH_4_^+^ by K^+^ ions was detected, consistent with the observation of a single G-quadruplex species in the ^1^H NMR spectrum.

### 2.7. Dimerization of [d(G_4_C_2_)]_4_ G-Quadruplex Structures Occurs through Stacking of 5-Membered Rings

Due to the high symmetry of the [d(G_4_C_2_)]_4_ G-quadruplex dimer in the presence of K^+^ ions, it was not possible to differentiate between restraints that corresponded solely to the stacking interface between two tetramolecular G-quadruplexes. Therefore, the high-resolution structure was calculated only for a monomeric unit (Appendix A). The resulting structure, which satisfied all experimentally derived structural restraints, closely resembled the one observed in the presence of NH_4_^+^ ions (PDB ID 8C7A, Figure 3B). However, in contrast to the structure in the presence of NH_4_^+^ ions, the C6 and C5 residues exhibited partial overlap, as supported by the C5H6–C6H6 cross-peak in the NOESY spectrum (Appendix A).

To evaluate the geometry at the stacking interface between the two tetramolecular [d(G_4_C_2_)]_4_ G-quadruplexes, four rotamers with different modes of the G-quartet stacking interface (stacking through 5-membered rings, 5/6-membered rings, 6-membered rings, and partial 6-membered rings, Appendix A) were constructed and subjected to MD simulation. A 200 ns MD simulation demonstrated that all four rotamers converged to a similar geometry, with the interface G-quartets composed of G1 residues stacking through 5-membered rings at a separation distance of 3.57 Å (Appendix A). Interestingly, the G-quartets at the interface displayed a smaller twist angle compared to the G-quartets composed of G1 and G2 residues (14.8° vs. 21.6°, Figure 3B).

### 2.8. K^+^ Ion between Interface G-Quartets Is Firmly Bound

To determine the preferential cation binding sites within the [d(G_4_C_2_)]_4_ G-quadruplex dimer, a G-quadruplex composed of two [d(G_4_C_2_)]_4_ G-quadruplexes was constructed, with K^+^ ions absent at the binding site O_3_ and an additional K^+^ ion positioned between the interface G-quartets comprised of G1 residues, denoted as O_GG_. Within a few picoseconds, it was observed that the K^+^ ion initially located at the binding site I migrated to the binding site O_3_ (Figure 3B). This movement could be attributed to the strong electrostatic repulsion between the K^+^ ion at the binding site I and the K^+^ ions at the binding sites O5 and O_GG_, with a distance of approximately 7.3 Å between the K^+^ ions at the binding sites I and O_GG_. MD simulation of the [d(G_4_C_2_)]_4_ G-quadruplex dimer confirmed that the K^+^ ion at the binding site O_GG_ remained firmly bound throughout the simulation. Thus, the occupancy of the binding site O_GG_ by a K^+^ ion is likely crucial for the dimerization of the tetramolecular G-quadruplexes.

## 3. Discussion

Tetramolecular G-quadruplexes are considered the simplest structures of G-quadruplex nucleic acids, where all strands align parallel to each other, and guanine residues adopt an *anti*-conformation around the glycosidic torsion angle [2,52]. Previous studies have suggested that both d(G_4_C_2_) and r(G_4_C_2_) can form such structures [53]. The interaction of cations with G-quadruplexes plays a significant role in their final structure formation. In our research, we employed ^15^NH_4_^+^ ions as substitutes for biologically relevant K^+^ ions and utilized NMR spectroscopy to investigate cation interactions with d(G_4_C_2_). Our focus was on identifying any additional cation binding sites beyond the G-quartets and vacant cation binding sites, as they can impact the overall structural stability, cation dynamics, and the directionality of cation exchange through the central G-quadruplex cavity. The presence of ^15^NH_4_^+^ ions induced the formation of a parallel tetramolecular [d(G_4_C_2_)]_4_ G-quadruplex with an all-*syn* G-quartet located at the 5′-end, and its high-resolution structure was determined using NMR. Previously, the formation of an all-*syn* G-quartet at the 5′-end has been observed only in the minor species of other tetramolecular G-quadruplexes, such as d(TG_3_T), d(TG_4_T), d(TG_5_T), and d(TG_8_T) [45,48]. Additionally, the [d(G_4_C_2_)]_4_ G-quadruplex structure is characterized by the presence of a C-quartet formed by C5 residues stacked over the neighboring G-quartet composed of G4 residues. Various structural motifs, including U-, A-, T-, and C-quartets, as well as base triads, enhance the thermodynamic stability and functional potential of G-quadruplexes, serving as recognition points for drugs or binding sites for biologically important molecules [54].

In the presence of ^15^NH_4_^+^ ions, the tetramolecular [d(G_4_C_2_)]_4_ G-quadruplex demonstrated the presence of three cation binding sites with different binding affinities, as evidenced by their uneven occupancy. The O_5_ binding site was fully occupied, while the other two sites (I and O_3_) were only half populated by cations. Our findings revealed that the most populated binding site, O_5_, within the [d(G_4_C_2_)]_4_ G-quadruplex is characterized by four *syn–anti* steps of guanine residues and is located between G-quartets with minimal discrepancy in the alignment of their negative charge centers, influencing the octahedral coordination geometries of cations. MD simulations indicated an additional binding site for ammonium ions in the cavity between the G- and C-quartets, composed of G4 and C5 residues. Stacking of the C-quartet to the G4-quartet is similar to the stacking of C-quartet to the G-quartet in the unimolecular G-quadruplex formed by the d[(G_4_C_2_)_3_G_4_] sequence [19]. The C-quartet mediated by amino-O2 hydrogen bonds was also observed in a parallel G-quadruplex formed by the d(TG_3_CG_2_T) from the SV40 virus [55] and the tetramolecular d(TG_2_CG_2_C) G-quadruplex associated with Fragile X syndrome [56]. This observation aligns with previous reports, where ion binding was observed not only between G-quartets but also to a C-quartet and between the G-quartet and loop residues [40,53,57,58,59].

Cations within G-quadruplexes display dynamic behavior, moving between binding sites and the bulk solution [42,60]. Previous studies have demonstrated that ^15^NH_4_^+^ ions exhibit faster motion within tetramolecular structures, predominantly parallel structures defined by an *anti*-orientation around the glycosidic bonds of all guanine residues within G-quartets, compared to monomolecular and bimolecular structures with parallel/antiparallel strand orientation and mixed *anti/syn*-conformations of guanine residues [42,46,60]. Hence, we propose that the molecularity of the G-quadruplex plays a critical role in dictating the speed of cation movement through the central channel. Additionally, the movement of cations towards the bulk solution and along the central cation cavity of G-quadruplexes can also be influenced by loop residues [44,46]. Consequently, the kinetics of cation motion appears to be closely associated with the structural rigidity of the G-quadruplex. However, the findings presented here, combining NMR data with molecular dynamics simulations, suggest that the rate of cation exchange is likely strongly influenced by the availability of vacant cation binding sites. The presence of unoccupied binding sites reduces the mutual repulsion between ions, which is the primary driving force behind cation movement through the central channel, in comparison to a scenario where all binding sites are occupied by cations.

The presence of an all-*syn* G-quartet composed of G1 residues in the central channel of the d(G_4_C_2_) G-quadruplex structure results in its closure at the 5′-end for the exchange of cations such as ^15^NH_4_^+^, while protons can still exchange between the interior and bulk solution. Consequently, the exchange of ^15^NH_4_^+^ ions within the G-quadruplex and the bulk solution takes place exclusively through the 3′-end of the structure. The observed asymmetrical movement of ^15^NH_4_^+^ ions within the [d(G_4_C_2_)]_4_ G-quadruplex, attributed to the *syn–anti* step, is supported by a significantly different energy barrier for ^15^NH_4_^+^ ions to cross the outer G-quartets composed of G1 and G4 residues. In contrast, the major form of the tetramolecular [d(TG_4_T)]_4_ G-quadruplex, which only exhibits *anti–anti* base stacking, results in essentially symmetrical ^15^NH_4_^+^ ion movement within the central channel, characterized by an average energy barrier of approximately 14 kcal/mol [61].

It is known that short oligonucleotide sequences with a single G-tract and lacking 5′- or 3′-end overhangs can form higher-order structures through the stacking of individual G-quadruplexes or interlocking of slipped G-rich strands from different G-quadruplexes [62,63,64,65]. Studies on the oligonucleotide d(TG_4_T) have demonstrated that replacing T with U at the 5′-end promotes the dimerization of two tetramolecular G-quadruplex units in the presence of ^15^NH_4_^+^ and K^+^ ions, but not Na^+^ ions [60]. Similar observations have been reported for the dimerization of the dimeric G-quadruplex formed by d(GCG_2_AG_4_AG_2_) via tail-to-tail stacking [66]. In our study, we showed that while ^15^NH_4_^+^ ions promoted the formation of tetramolecular G-quadruplexes, the presence of K^+^ ions induced higher-ordered species where two tetramolecular [d(G_4_C_2_)]_4_ G-quadruplexes stack. Titrating KCl into a solution of [d(G_4_C_2_)]_4_ folded in the presence of ^15^NH_4_Cl revealed that a single K^+^ ion at the interface of two separate tetramolecular G-quadruplex structures is sufficient to initiate dimerization. Recent studies have also indicated that at 100 mM of KCl, d(G_4_C_2_)_n_ sequences with n equal to 1, 2, and 4 form supramolecular G-quadruplex assemblies, known as G-wires, extending beyond 80 nm in length [67].

Hexanucleotide repeat expansions of d(G_4_C_2_) in the *c9orf72* gene are the most common genetic cause of ALS and FTLD [68]. Although the full extent and implications of G-quadruplex formation in this region are yet to be determined, it is known that the transcription of the *c9orf72* gene is negatively regulated by d(G_4_C_2_)_n_ G-quadruplexes formed within the gene [20]. It is speculated that the formation of r(G_4_C_2_)_n_ prevents protein–RNA interactions crucial for the pathomechanism leading to ALS and FTLD [69,70]. Recent studies have also indicated that the stabilization of d(G_4_C_2_)_n_ quadruplexes by small molecules mitigates ALS/FTLD pathology both in vitro and in vivo [71,72]. Our study highlighted that the formation of an all-*syn* G-quartet may exhibit different binding properties compared to an *anti*-quartet, which should be considered when designing small molecules as potential drugs. Additionally, we demonstrated that K^+^ ions, which are abundant in the intracellular environment, significantly stabilized the [d(G_4_C_2_)]_4_ G-quadruplex compared to other ions and induced its immediate dimerization. The formation of such higher-ordered structures mediated by cations in vivo could introduce additional structural features in the hexanucleotide repeat expansions of the *c9orf72* gene and potentially impact genomic stability and the pathophysiology of ALS and FTLD.

## 4. Materials and Methods

### 4.1. Sample Preparation

Oligonucleotide d(G_4_C_2_) was synthesized on K&A Laborgeraete GbR DNA/RNA Synthesizer H-8 using standard phosphoramidite chemistry in DMT-off mode. Deprotection was performed with overnight incubation in 20% aqueous ammonia at 50 °C. Then, 2 M LiCl was added before heating samples at 90 °C for 10 min. Samples were purified and concentrated using an ultra-filtration device (Merck Millipore, Watford, UK) and an ultra-filtration membrane with a cut-off mass of 1 kDa (regenerated cellulose, Millipore). The samples were lyophilized overnight and diluted in 90% H_2_O and 10% ^2^H_2_O to a final concentration of 2.0 mM per strand in the presence of 100 mM of ^15^NH_4_Cl or 100 mM of ^15^NH_4_Cl and 80 mM of KCl. Sample concentrations were determined by measuring absorbance at 260 nm using a Varian Cary 100 Bio UV-VIS spectrometer (Varian Inc., Palo Alto, CA, USA, Agilent Tecnologies, Santa Clara, CA, USA). The extinction coefficient was determined by the nearest-neighbor method.

### 4.2. Circular Dichroism Spectroscopy

All CD spectra were recorded on an Applied Photophysics Chirascan CD spectrometer (Leatherhead, UK) at 25 °C using a 0.1 or 1 mm-path-length quartz cell. The wavelength ranged from 220 to 320 nm. For CD spectra, an aliquot of the NMR sample was taken. Then, 100 mM ^15^NH_4_Cl or 80 mM KCl and 100 mM ^15^NH_4_Cl solutions were used for baseline correction. For CD melting experiments, 70 μM oligonucleotide solutions with 10 mM of Li-cacodylate buffer pH 6.0 and either 100 mM of NH_4_Cl or 100 mM of NH_4_Cl and 80 mM of KCl were used. A mixture of 10 mM of Li-cacodylate buffer and the respective salt concentrations were used as blank samples. Unfolding/folding processes were followed between 10 and 95 °C at a heating rate of 0.1 °C min^−1^. CD spectra were acquired every 1 °C. Transition temperatures (T_m_) were calculated as an average of T_m_, determined using the first derivative method at wavelengths of 260 and 300 nm.

### 4.3. UV Spectroscopy

UV melting experiments were performed on a Varian Cary 100 Bio UV-VIS spectrometer (Varian Inc., Agilent Tecnologies, Santa Clara, CA, USA) equipped with a thermoelectric temperature controller. Samples were prepared at 300, 100, 30, and 10 μM oligonucleotide concentrations in 10 mM of Li-cacodylate buffer, pH 6.0, 100 mM of NH_4_Cl, or 10 mM of Li-cacodylate buffer, pH 6.0, 100 mM of NH_4_Cl and 80 mM of KCl. Unfolding/folding processes were followed between 10 and 95 °C at a heating rate of 0.1 °C min^−1^. Absorbance at 295 nm was measured every 0.5 °C. Then, 1, 2, 5, and 10 mm-path-length quartz cells were used for 300, 100, 30, and 10 μM samples, respectively. A combination of mineral oil and a fixed cuvette cap was used to prevent evaporation and sample loss due to high temperatures. A stream of nitrogen was applied throughout the measurements to prevent condensation at lower temperatures. Transition temperatures (T_m_) were determined using the first derivative method.

### 4.4. NMR Spectroscopy

NMR experiments were performed on Agilent Technologies (USA) DD2 600 MHz and VNMRS 800 MHz NMR spectrometers at 25 °C (unless otherwise stated) using a triple-resonance cold probe. The 1D ^1^H spectra were acquired with the use of DPFGSE solvent suppression. Diffusion coefficient measurements were performed by a spin-echo pulse sequence with PFG gradient strengths between 0.49 and 29.06 G cm^−1^. The 2D NOESY spectra were acquired with mixing times ranging from 80 to 250 ms with the use of DPFGSE water suppression. Imino proton resonances were assigned through cross-peak analysis in the imino-imino and imino-aromatic regions of 2D NOESY spectra (Appendix A). The assignment of non-exchangeable aromatic and sugar proton resonances was accomplished by continuous sequential walks from G1 to C6 in the H8/H6-H1′, H8/H6-H2′/H2″, and H8/H6-H3′ regions of 2D NOESY spectra (80, 150, and 250 ms mixing times) (Appendix A), complemented with 2D DQF-COSY, 2D TOCSY (20 and 80 ms mixing times), 2D ROESY (80 ms mixing time), and 2D ^1^H-^31^P COSY spectra (Appendix A), acquired on samples in 100% ^2^H_2_O. Resonances of exchangeable protons were assigned using 2D NOESY recorded in 90% H_2_O and 10% ^2^H_2_O. The number and location of ^15^NH_4_^+^ ion binding sites were determined using 2D ^1^H-^15^N HSQC, 2D NOESY, and 2D ROESY spectra. Assignment of well-resolved autocorrelation cross-peaks in the HSQC spectrum corresponding to bound ^15^NH_4_^+^ ions was established through NOE and ROE connectivity between ^15^NH_4_^+^ ion protons and neighboring imino protons of guanine residues. The movements of ^15^NH_4_^+^ ions and protons were followed by a series of ^1^H-^15^N NzExHSQC and ^1^H-^15^N HzExHSQC spectra with mixing times ranging from 13 ms to 3 s, and from 14 ms to 2 s, respectively. NMR spectra were processed and analyzed using VNMRJ 3.2 (Varian Inc.), Sparky (UCSF, San Francisco, CA, USA), and Mestrenova 9.1 software (Mestrelab Research, Santiago de Compostela, Spain).

### 4.5. ^1^H-^15^N NzExHSQC Data Analysis

Volumes of autocorrelation and cross-peaks in ^1^H-^15^N NzExHSQC spectra were integrated using VNMRJ 3.2 (Varian Inc.) software. The volume of autocorrelation peak O_5_ at a mixing time of 13 ms was set to 100, and all other integrals were normalized to this value. Exchange rates were determined using iterative least-squares fitting, performed in Origin 8.1 (OriginLab, Northampton, MA, USA). The decrease of intensity of the autocorrelation peaks B, O_5_, I, and O_3_, as a function of the mixing time (*τ_m_*), is a result of ion movement from the original binding site as well as the spin-lattice relaxation (*T*_1_), and is best described by the bi-exponential function [44,46]:(1)$$V_{auto}\left(\tau_{m}\right)=A_{1}\left[e^{-r_{1}\tau_{m}}\right]+A_{2}\left[e^{-\frac{\tau_{m}}{T_{1}}}\right]$$
where *A*_1_ and *A*_2_ are scaling factors and *r*_1_ is a rate constant. Estimated *T*_1_ relaxation times of autocorrelation peaks B and I were used in the analysis of cross-peak volumes as a function of the mixing time, corresponding to ^15^NH_4_^+^ ions exchanging with the bulk solution and ions moving within the [d(G_4_C_2_)]_4_ G-quadruplex, respectively. Cross-peak volumes as a function of the mixing time were fitted to the following equation [44,46]:(2)$$V_{cross-peak}\left(\tau_{m}\right)=A[e^{-\frac{\tau_{m}}{T_{1}}}(1-e^{-k\tau_{m}})]$$
where *A* is a scaling factor and *k* is the exchange rate.

### 4.6. NMR Restraints and Structure Calculations

NOE distance restraints for exchangeable and non-exchangeable protons were obtained from 2D NOESY spectra with a mixing time of 150 ms, recorded at 25 °C in 90% H_2_O, 10% ^2^H_2_O, and 100% ^2^H_2_O, respectively. Due to severe overlap of cytosine H5-H6 cross-peaks, the average volume of the H1′-H2″ cross-peak was used as a distance reference of 2.20 Å [73]. NOE interactions were classified as strong (1.8–3.6 Å), medium (2.6–5.0 Å), and weak (3.5–6.5 Å). The torsion angle, χ, was restrained between 30 and 90° for residues in *syn*-orientation and between 200 and 280° for residues in *anti*-orientation. Due to the symmetric nature of G-quadruplexes formed by d(G_4_C_2_) in the presence of ^15^NH_4_^+^ or K^+^ ions, NOE connectivity of only one strand was observed in NOESY spectra. In order to calculate high-resolution structures of G-quadruplexes adopted by d(G_4_C_2_), all NMR-derived restraints were multiplied four times and assigned to chains A, B, C, and D. Structure calculations were performed with AMBER 14 software using the parmbsc0 force field [74], with parmχOL4 [75] and parmε/ζOL1 [76] modifications. The initial extended structure was obtained using the leap module of AMBER 14. Then, 1000 structures were calculated in 160 ps NMR-restrained simulated annealing (SA) simulations using the generalized Born-implicit model. The cut-off for non-bonded interactions was 999 Å, and the SHAKE algorithm for hydrogen atoms with tolerance of 0.0005 Å was used. Random velocities were used for each SA simulation. In the first 20 ps of simulation, the temperature was increased from 300 to 1000 K. In the next 20 ps, the temperature was kept constant at 1000 K. Then, molecules were gradually cooled, first to 500 K in the next 50 ps of SA, and then to 100 K in the next 30 ps of SA. In the last 40 ps of SA, the temperature was scaled to 0 K. This long cooling process was used to prevent stacking of cytosines in grooves of the G-quadruplex core. Force constants for hydrogen bond restraints and NOE-derived distance restraints were 20 kcal mol^−1^ Å^−2^, while 200 kcal mol^−1^ rad^−2^ was used for the torsion angle χ, and 20 kcal mol^−1^ rad^−2^ for torsion angles ε and β. No planarity restraints for G-quartets were used during structure calculation. Structures were energy-minimized with a maximum of 20,000 steps, and 10 structures were selected based on the smallest number of restraint violations and the lowest energy. UCSF Chimerasoftware version 1.17.1 was used for visualization and preparation of figures [77]. Atomic coordinates and the list of chemical shifts for [d(G_4_C_2_)]_4_:^15^NH_4_ have been deposited in the Protein Data Bank under accession numbers PDB ID 8C7B and BMRB ID 30743. Atomic coordinates and the list of chemical shifts for [d(G_4_C_2_)]_4_:K^+^ have been deposited in the Protein Data Bank under accession numbers PDB ID 8C7A and BMRB ID 30914.

### 4.7. Molecular Dynamics Simulation

Initially, three ammonium or potassium ions were placed between the O_5_, I, or O_3_ binding sites, respectively. Three additional models were built, in which the ammonium ion was absent in the O_5_, I, or O_3_ binding sites. The [d(G_4_C_2_)]_4_ G-quadruplex dimer models were built based on the last snapshots of the MD simulation of the monomer. The dimeric model of the [d(G_4_C_2_)]_4_ G-quadruplex in the presence of potassium ions was built based on the stacking of the G-quartet composed of G1 residues. Four different models were generated by manual rotation of one subunit along the axis, defined by the center of the G-quartet to produce four different types of dimer-stacked interfaces, i.e., partial 6-membered rings, 6-membered rings, 5/6-membered rings, and 5-membered rings (Appendix A), as observed in crystallographic structures [78]. A K^+^ ion was manually positioned between the dimer interfaces. The G-quadruplexes were then placed in a truncated octahedral (for monomers) or cubic (for dimers) box of TIP3P water molecules, with the box border at least 10 Å away from any atoms of the G-quadruplex. Extra ammonium or potassium ions were added to neutralize the negative charges of the G-quadruplex. After the models were built, the ion positions were randomized using CPPTRAJ [79] by swapping random water and ion positions, such that no ion was closer than 4 Å to another and all ions were greater than 6 Å away from the DNA, to avoid any biasing created by the initial placement of the ions.

The force field for MD simulation was the same as the SA simulation. The simulations were performed with the CUDA version of the pmemd module of AMBER 14 [80,81,82,83]. Before the MD simulation, the systems were subjected to a series of minimizations and equilibration. The equilibration protocol started with 500 steps of steepest descent minimization, followed by 500 steps of conjugate gradient minimization with 25 kcal mol^−1^ Å^−2^ position restraints on DNA atoms. Then, the system was heated from 0 to 300 K during 100 ps, with position restraints of 25 kcal mol^−1^ Å^−2^ on the G-quadruplex. Afterwards, the system underwent minimization with 5 kcal mol^−1^ Å^−2^ restraints on DNA atoms using 500 steps of the steepest descent method, followed by 500 steps of the conjugate gradient. Then, the system was equilibrated during 50 ps at a constant temperature of 300 K and pressure of 1 atm with 5 kcal mol^−1^ Å^−2^ position restraints on DNA atoms. An analogous series of alternating minimizations and equilibrations was performed, consecutively using decreasing position restraints of 4, 3, 2, and 1 kcal mol^−1^ Å^−2^. Finally, an equilibration using position restraints of 0.5 kcal mol^−1^ Å^−2^ and starting velocities from the previous equilibration, followed by a short 50 ps molecular dynamics, took place without any restraints. Pressure coupling used during equilibration was set to 0.2, and coupling during the last molecular dynamics phase was set to 5.

The production simulation was carried out at a constant pressure of 1 atm, and a constant temperature of 300 K, maintained using Langevin dynamics with a collision frequency of 2.0, respectively. Periodic boundary conditions were used, and electrostatic interactions were calculated by the particle mesh Ewald method [84,85], with the non-bonded cut-off set to 9 Å. The SHAKE algorithm [86] was applied to bonds involving hydrogens, and a 2 fs integration step was used. The production run was continuously carried out for 200 ns, and the snapshots were written at every 1 ps. Trajectories were analyzed using the CPPTRAJ module of AMBER.

The potential of mean force (PMF) of the movement of NH_4_^+^ ions inside the channel of the [d(G_4_C_2_)]_4_ G-quadruplex was obtained using the adaptive steered molecular dynamics (ASMD) method [50,51]. The G-quadruplex central channel axis was used as reaction coordinates, which started from the mass center of quartets consisting of C5 and G4 residues to the outside of the quartet composed of G1 residues. The reaction coordinate was divided into 14 stages. In each stage, the NH_4_^+^ ion was pulled for 1 Å in 0.5 ns, and 100 trajectories were generated, with the velocities for each trajectory distributed based on random number sequences. The coordinates of the [d(G_4_C_2_)]_4_ G-quadruplex for ASMD simulation were taken from the last snapshots of the 200 ns regular MD simulation. The simulations were performed in the NVT ensemble at 300 K. Finally, the potential of mean force (PMF) was calculated in each stage using Jarzynski’s equality [87].

## 5. Conclusions

Cations play a significant role in influencing the formation and thermal stability of G-quadruplexes. In this study, we demonstrated that the hexanucleotide repeat d(G_4_C_2_), which is associated with ALS and FTLD diseases, formed a tetramolecular G-quadruplex unit in the presence of ^15^NH_4_^+^ ions. Furthermore, the introduction of biologically relevant K^+^ ions promoted immediate dimerization of two such units at the 5′-ends, resulting in a species with higher thermal stability. By utilizing ^15^NH_4_^+^ ions and NMR spectroscopy, we have uncovered that the specific nature of cations positioned between G-quartets within the G-quadruplex units is not crucial for the formation of higher-order structures. Instead, it is the presence of K^+^ ions at the interface between the separate G-quadruplex units that drives their dimerization. These higher-ordered structures, facilitated by cations, could potentially impact genomic stability, and contribute to the pathophysiology of ALS and FTLD. An important structural feature of the resulting tetramolecular G-quadruplex structure is the presence of an all-*syn* G-quartet at the 5′-end. This unique feature has the potential to influence the binding properties of small molecules, including potential drugs, and therefore, it has implications for their design. Another characteristic of the G-quadruplex formed by d(G_4_C_2_) is the slow dynamics of ^15^NH_4_^+^ ion movements throughout the central channel. Additionally, the exchange of cations into the bulk solution occurred exclusively at the 3′-end. These observations are a consequence of the all-*syn* G-quartet at the 5′-end of the G-quadruplex, as well as the presence of vacant cation binding sites. Understanding such cation movements can also find applications in the design of nanostructures with specific properties.

## Figures and Tables

**Figure 1 ijms-24-13437-f001:**
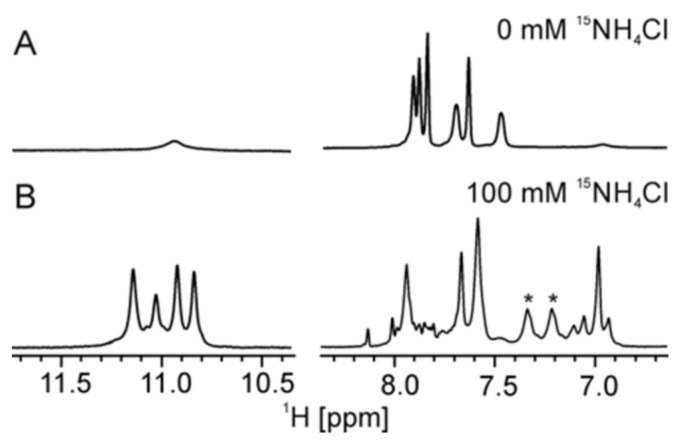
Imino and aromatic regions of the ^1^H NMR spectrum of d(G_4_C_2_) in the absence (**A**) and the presence (**B**) of 100 mM of ^15^NH_4_Cl. Spectra were recorded in 90% H_2_O and 10% ^2^H_2_O at 25 °C, pH 6.0, on a 600 MHz NMR spectrometer. The oligonucleotide concentration was 2.0 mM per strand. The doublet signal (J = 74 Hz) denoted with asterisks corresponds to ammonium ions bound to binding site O_5_ within the G-quadruplex.

**Figure 2 ijms-24-13437-f002:**
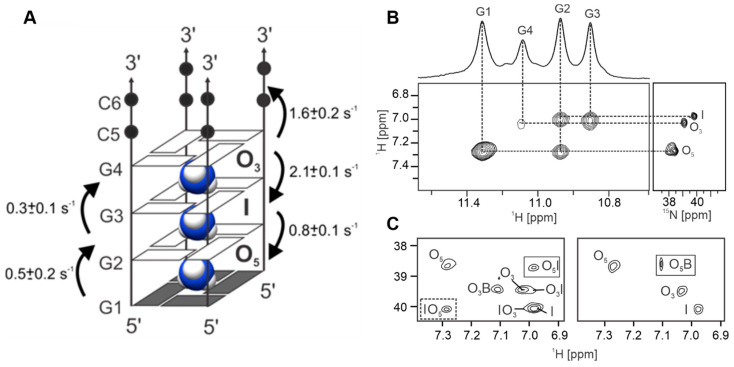
(**A**) Folding topology and cation binding sites within G-quadruplex adopted by d(G_4_C_2_) in the presence of ^15^NH_4_^+^ ions, with the three ion binding sites shown. *Syn*- and *anti*-conformations of guanines are represented with grey and white rectangles, respectively. The arrows represent the observed movements between cation binding sites, with apparent rate constants depicted next to them. (**B**) Imino-aromatic region of the NOESY spectrum (left), showing cross-peaks between bound ^15^NH_4_^+^ ions and nearby imino protons, and the 2D ^1^H-^15^N HSQC spectrum (right) exhibiting cross-peaks corresponding to ^15^NH_4_^+^ ions in different chemical environments within [d(G_4_C_2_)]_4_. The vertical and horizontal dashed lines connect resonances of guanine imino and ^15^NH_4_^+^ ion protons. (**C**) The 2D ^1^H-^15^N NzExHSQC spectrum at a mixing time (τ_m_) of 1.1 s (**left**) and the 2D ^1^H-^15^N HzExHSQC spectrum at a mixing time (τ_m_) of 40 ms (**right**). Annotations of autocorrelation and exchange cross-peaks are shown next to the cross-peaks.

**Figure 3 ijms-24-13437-f003:**
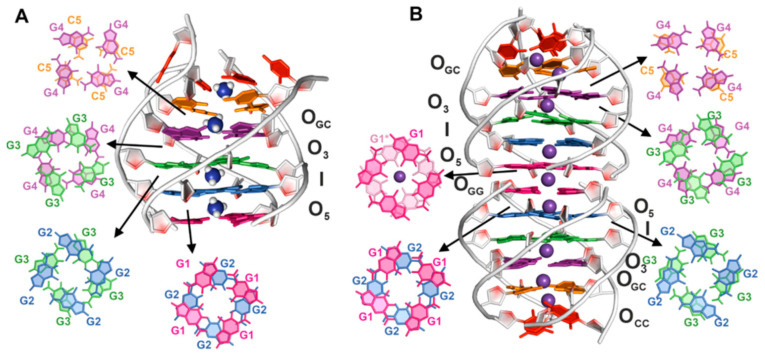
(**A**) Lowest energy structure of the [d(G_4_C_2_)]_4_ G-quadruplex in the presence of ^15^NH_4_^+^ ions (PDB ID 8C7B) and (**B**) of the dimeric form of the [d(G_4_C_2_)]_4_ G-quadruplex in the presence of K^+^ ions (PDB ID 8C7A). Residues G1, G2, G3, G4, C5, and C6 are shown in magenta, blue, green, violet, orange, and red, respectively. Cations within both structures are shown. Ion binding sites in (**A**) and (**B**) are marked with O_5_, I, O_3_, and O_GC_. O_GC_ is the binding site between G4 and C5 residues. In (**B**) an additional binding site, O_GG_, is present. It is located between G1 and G1 residues, formed by 5′ stacking of two [d(G_4_C_2_)]_4_ G-quadruplexes. Stacking of G-quartets and G-quartets and cytosines are shown.

**Figure 4 ijms-24-13437-f004:**
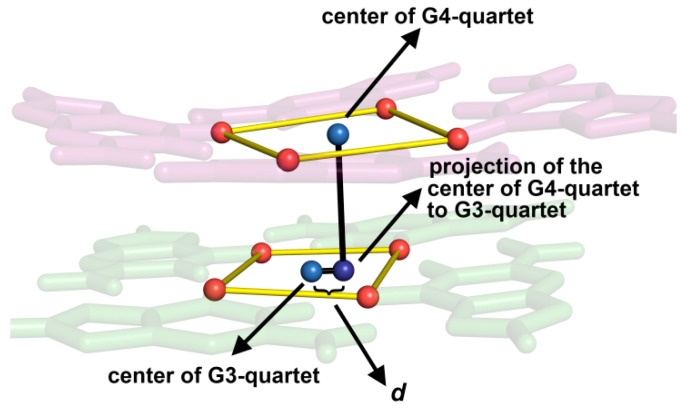
The projection of the center of a G-quartet composed of G4 residues (marked as G4-quartet) to the adjacent G-quartet composed of G3 residues (marked as G3-quartet) within [d(G_4_C_2_)]_4_ in the presence of ^15^NH_4_^+^ ions. Due to clarity, the ^15^NH_4_^+^ ion between G-quartets is not present. The O6 atoms of the G-quartets are shown as red sphere models, and centers of the G-quartets are shown in blue. d is the distance between the centers of two adjacent G-quartets.

**Figure 5 ijms-24-13437-f005:**
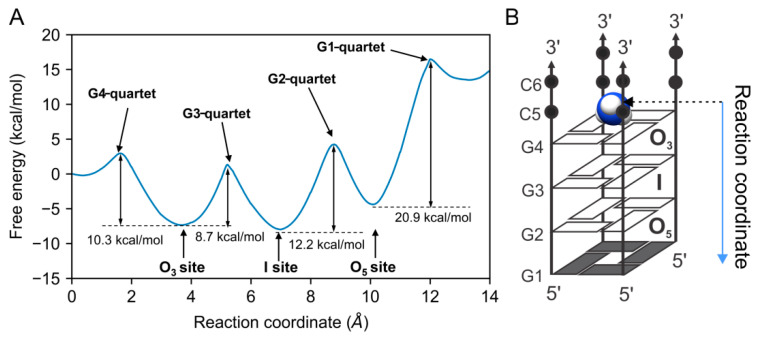
(**A**) Free-energy profile for NH_4_^+^ ion movements through G-quartets of [d(G_4_C_2_)]_4_ G-quadruplex and the pulling direction of ^15^NH_4_^+^ ions, as well as (**B**) the representation of the reaction coordinate. G-quartets composed of G1, G2, G3, and G4 residues are marked as G1-, G2-, G3-, and G4-quartets, respectively.

**Figure 6 ijms-24-13437-f006:**
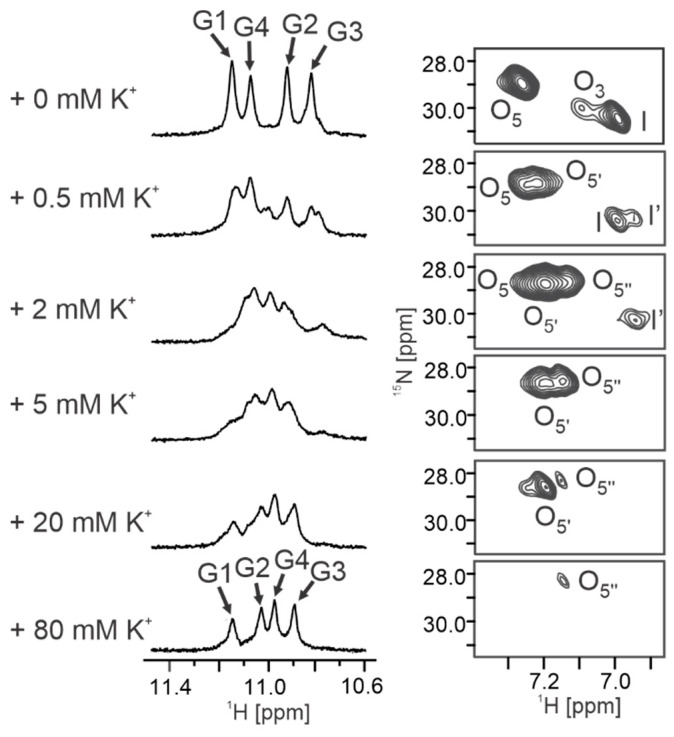
Imino regions of ^1^H NMR and corresponding ^1^H-^15^N HSQC spectra of [d(G_4_C_2_)]_4_ at 25 °C, pH 6.0, in 10% ^2^H_2_O in the presence of 100 mM of ^15^NH_4_Cl and gradual addition of KCl. The oligonucleotide concentration was 2.0 mM per strand. Assignments of imino resonances at the beginning and end of titration are indicated above the signals. Annotations of cross-peaks are shown next to the peaks.

**Figure 7 ijms-24-13437-f007:**
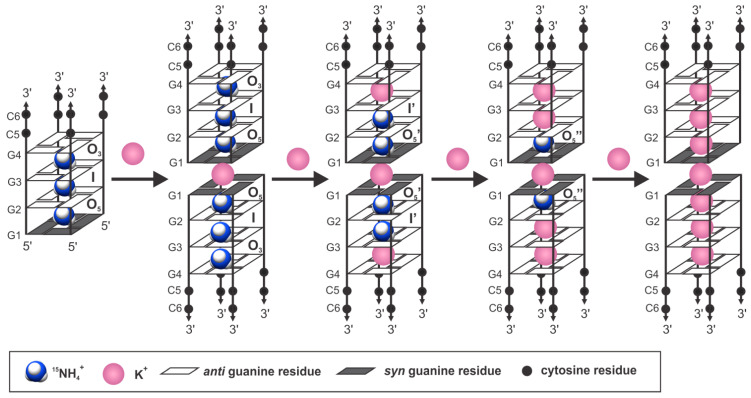
Gradual exchange of ^15^NH_4_^+^ by K^+^ ions and the formation of dimeric species as the concentration of K^+^ ions is increased from 0 to 80 mM.

## Data Availability

Atomic coordinates and the list of chemical shifts for [d(G_4_C_2_)]_4_:^15^NH_4_ have been deposited in the Protein Data Bank under accession numbers PDB ID 8C7B and BMRB ID 30743. Atomic coordinates and the list of chemical shifts for [d(G_4_C_2_)]_4_:K^+^ have been deposited in the Protein Data Bank under accession numbers PDB ID 8C7A and BMRB ID 30914.

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
