# Peer review of "Insight into Tetramolecular DNA G-Quadruplexes Associated with ALS and FTLD: Cation Interactions and Formation of Higher-Ordered Structure"

_ijms, 2023, doi:10.3390/ijms241713437_

Round 1

Reviewer 1 Report

In this study, the authors delve into the G4C2 hexanucleotide repeat expansion in the c9orf72 gene, which is known to be a significant cause of familial ALS and FTLD. The research specifically focuses on the relationship between the G4C2 repeat and G-quadruplexes. By employing nuclear magnetic resonance spectroscopy and molecular dynamics simulations, the researchers have successfully investigated cation binding and exchange dynamics. Their findings reveal the formation of a tetramolecular G-quadruplex structure with an all-syn quartet at the 5'-end. Furthermore, the study demonstrates that the introduction of K+ ions leads to the stacking of G-quadruplexes via their 5'-end G-quartets, resulting in the formation of stable higher-ordered species. The work provides valuable insights into the G-quadruplex structure formed by G4C2 DNA and sheds light on the impact of biologically relevant potassium ions on this structure.

Major comments:

1.       The authors focus on investigating the tetramolecular G-quadruplex structure formed by single G4C2 DNA units, even though the G4C2 repeat in C9orf72-ALS patients can be much longer, ranging from hundreds to thousands of repeats. These longer repeats are more likely to form intramolecular DNA or RNA G-quadruplexes. It would be beneficial for the study to address why the authors chose to explore tetramolecular G4 structures instead of intramolecular G4 structures, which are more relevant to what occurs in cells.

2.       The NMR assays conducted to characterize the G-quadruplex structure are well-performed. To further validate the formed G-quadruplex structures, it might be worth considering performing similar NMR assays using control sequences, such as mutated G4C2 sequences designed to disrupt the G-track to inhibit G-quadruplex formation.

Minor comments:

1.       Considering that the study primarily focuses on G4C2 repeat DNA, the title should be revised to "DNA G-quadruplex" instead of just "G-quadruplex".

2.       In the introduction section, the authors may consider providing additional biological background information on the C9orf72 repeat DNA and RNA, along with the cellular regulators involved, such as DAXX, DHX36, and hnRNP H. (References: PMID: 36822200, PMID: 33855846, PMID: 34174288, PMID: 27623008). This additional context will enhance the reader's understanding of the subject matter.

Author Response

Overview of the reviewer: In this study, the authors delve into the G4C2 hexanucleotide repeat expansion in the c9orf72 gene, which is known to be a significant cause of familial ALS and FTLD. The research specifically focuses on the relationship between the G4C2 repeat and G-quadruplexes. By employing nuclear magnetic resonance spectroscopy and molecular dynamics simulations, the researchers have successfully investigated cation binding and exchange dynamics. Their findings reveal the formation of a tetramolecular G-quadruplex structure with an all-syn quartet at the 5'-end. Furthermore, the study demonstrates that the introduction of K+ ions leads to the stacking of G-quadruplexes via their 5'-end G-quartets, resulting in the formation of stable higher-ordered species. The work provides valuable insights into the G-quadruplex structure formed by G4C2 DNA and sheds light on the impact of biologically relevant potassium ions on this structure.

Our response: We are grateful for the constructive feedback and the suggestions on how to increase the impact of the manuscript.

Major comments:

Comment 1: The authors focus on investigating the tetramolecular G-quadruplex structure formed by single G4C2 DNA units, even though the G4C2 repeat in C9orf72-ALS patients can be much longer, ranging from hundreds to thousands of repeats. These longer repeats are more likely to form intramolecular DNA or RNA G-quadruplexes. It would be beneficial for the study to address why the authors chose to explore tetramolecular G4 structures instead of intramolecular G4 structures, which are more relevant to what occurs in cells.

Our response: We agree that intramolecular DNA G-quadruplexes formed by longer (G4C2)n repeats are more likely to form in solution. However, such longer (G4C2)n repeats show high degree of structural polymorphism. For instance, even the shortest oligonucleotide model of G4C2 repeats with the ability to form a unimolecular G-quadruplex (G4C2)3G4 exists as an equilibrium between two major and several minor species. (https://www.ncbi.nlm.nih.gov/pmc/articles/PMC6265483/, ref. 19 in manuscript) These structures are very similar to each other with overlapping signals that are very difficult to assign to a specific species. High structural polymorphism of longer G4C2 repeats would therefore prevent detailed study of cation binding sites and their exchange. We have added an explanation of the sequence choice in the introduction.

Comment 2: The NMR assays conducted to characterize the G-quadruplex structure are well-performed. To further validate the formed G-quadruplex structures, it might be worth considering performing similar NMR assays using control sequences, such as mutated G4C2 sequences designed to disrupt the G-track to inhibit G-quadruplex formation.

Our response: Alongside the NMR assay we have also performed CD and UV-Vis measurements (Figures S8, S9 and S10) which were all consistent with the reported tetramolecular parallel structure determined by NMR. Additionally, the obtained structure is in line with previously published structures on similar sequences (TGGGGT (https://pubmed.ncbi.nlm.nih.gov/20735000/) ref. 60 in manuscript, TGGGT https://www.ncbi.nlm.nih.gov/pmc/articles/PMC3510487/ ref. 45 in manuscript). Furthermore, a recent study by Gong et al (https://www.pnas.org/doi/10.1073/pnas.2013230118, ref. 29 in manuscript) showed that even a single mutation in the G-tract prevents formation of G-quadruplexes. We therefore feel that while suggested additional study using mutagenesis would provide useful insight into the folding pathways of G-quadruplexes, this work falls out of the scope of this manuscript. In the introduction, we have added a statement that reflects these recent findings.

Minor comments:

Comment 3: Considering that the study primarily focuses on G4C2 repeat DNA, the title should be revised to "DNA G-quadruplex" instead of just "G-quadruplex".

Our response: We have amended the title according to reviewer’s suggestion.

Comment 4: In the introduction section, the authors may consider providing additional biological background information on the C9orf72 repeat DNA and RNA, along with the cellular regulators involved, such as DAXX, DHX36, and hnRNP H. (References: PMID: 36822200, PMID: 33855846, PMID: 34174288, PMID: 27623008). This additional context will enhance the reader's understanding of the subject matter.

Our response: According to the reviewer’s suggestion we have included a paragraph in the introduction to include additional biological background information on the C9orf72 repeat.

Reviewer 2 Report

The manuscript by Zalar et al. presents an interesting investigation into the G-quadruplex structures formed by G4C2 sequences, which are associated with repeat expansion in the c9orf72 gene. These and related sequences have been intensively studied in recent years due to their linkage with neurological disorders, most notably amyotrophic lateral sclerosis (ALS) and frontotemporal lobar degeneration (FTLD). Despite the numerous related reports, the findings elucidated herein remain novel and interesting for a broader readership.

The study is technically very well-conducted, and the paper is nicely written. In my opinion that the paper is ready for publication in its current state. Nevertheless, I believe in could be enhanced through the incorporation of some additional details concerning the C-tetrad. Previous instances of C-tetrads have been observed in the context of G-quadruplexes. It would be instructive to compare those tetrads with the instance under investigation in this work (adding the corresponding citations). Specifically, it would be interesting to clarify whether these tetrads are stabilized by direct hydrogen bonds (amino-O2), or if they are mediated by cations.

Author Response

Overview of the reviewer: The manuscript by Zalar et al. presents an interesting investigation into the G-quadruplex structures formed by G4C2 sequences, which are associated with repeat expansion in the c9orf72 gene. These and related sequences have been intensively studied in recent years due to their linkage with neurological disorders, most notably amyotrophic lateral sclerosis (ALS) and frontotemporal lobar degeneration (FTLD). Despite the numerous related reports, the findings elucidated herein remain novel and interesting for a broader readership.

The study is technically very well-conducted, and the paper is nicely written. In my opinion that the paper is ready for publication in its current state. Nevertheless, I believe in could be enhanced through the incorporation of some additional details concerning the C-tetrad. Previous instances of C-tetrads have been observed in the context of G-quadruplexes. It would be instructive to compare those tetrads with the instance under investigation in this work (adding the corresponding citations). Specifically, it would be interesting to clarify whether these tetrads are stabilized by direct hydrogen bonds (amino-O2), or if they are mediated by cations.

Our response: Thank you for taking the time to review our paper and your positive evaluation. The observed C-quartets are indeed mediated by the presence of hydrogen bonds in amino-O2 geometry further stabilized by the presence of the cations. We have added additional explanation into the results section. According to your suggestion, we have also added a short comparison on the C-quartet observed here to those reported in previously published literature to the discussion.

Round 2

Reviewer 1 Report

The authors have adequately addressed my concerns. No more comments.